# Effects of Water and Nitrogen on Grain Filling Characteristics, Canopy Microclimate with Chalkiness of Directly Seeded Rice

Yongjian Sun [1,*], Yunxia Wu [1], Yuanyuan Sun [2], Yinghan Luo [1], Changchun Guo [1], Bo Li [1], Feijie Li [1], Mengwen Xing [1], Zhiyuan Yang [1] and Jun Ma [1,*]

[1] Crop Ecophysiology and Cultivation Key Laboratory of Sichuan Province, Rice Research Institute, Sichuan Agricultural University, Chengdu 611130, China; Yuxia@stu.sicau.edu.cn (Y.W.); Yinghan@stu.sicau.edu.cn (Y.L.); changchuns1991@stu.sicau.edu.cn (C.G.); li742681930@stu.sicau.edu.cn (B.L.); Feijie@stu.sicau.edu.cn (F.L.); Mengwen@stu.sicau.edu.cn (M.X.); 14236@sicau.edu.cn (Z.Y.)

[2] Institute of Plateau Meteorology, China Meteorological Administration, Chengdu 610072, China; ytyy21@cma.cn

\* Correspondence: yongjians1980@sicau.edu.cn (Y.S.); majun@sicau.edu.cn (J.M.)

**Abstract:** In order to determine how to reduce the chalkiness of rice grains through irrigation modes and nitrogen (N) fertilizer management. The experiment was designed using three irrigation modes (flooding ($W_1$), dry–wet alternating ($W_2$), and dry alternating ($W_3$)), three N application strategies (under 150 kg ha$^{-1}$, the application ratio of base:tiller:panicle fertilizer (30%:50%:20% ($N_1$), 30%:30%:40% ($N_2$), and 30%:10%:60% ($N_3$)), and zero N as the control ($N_0$) in 2019 and 2020. The results revealed that water–nitrogen interactions had a significant or extremely significant effect on the chalkiness characteristics of the superior and inferior grains. Compared with $W_1$ and $W_3$ treatments, $W_2$ coupled with the $N_1$ application strategy can further optimize grain filling characteristics and canopy microclimate parameters, thereby reducing grain chalkiness. Correlation analysis revealed that increasing grain filling parameters ($G_{max}$ or $G_{mean}$) and mean grain filling rates (MGRs) during the mid-filling stage in superior grains of the primary branches and inferior grains of the secondary branches, which were important factors in water–nitrogen interaction effects, could further reduce chalkiness. Improving the canopy microclimate (daily average temperature difference and daily average light intensity difference) during the early-filling stage for inferior grains and the mid-filling stage for superior grains could be another important method to reduce chalkiness.

**Keywords:** the grain filling stage; canopy microclimate; grain position on panicle; rice chalkiness; water–nitrogen interaction





## 1. Introduction

Directly seeded rice production has a long history, and is primarily used because it results in a significantly shorter vegetative growth period and has differences in temperature and light utilization characteristics compared with traditional manual transplanting and machine transplanting [1,2]. In addition, this method can save seedlings and transplanting, saving labor and cost and facilitating stable yield; thus, this will become an inevitable trend in the development of simplified and efficient rice production in the future [2,3]. Socioeconomic development has resulted in a higher demand for quality rice, and high quality and high efficiency have become important directions for continuous research on rice production [4]. Grain chalkiness is a white and opaque part of rice endosperm, which is caused by the insufficient accumulation of starch and protein particles in endosperm [5,6]. It is easy to break during processing, resulting in a significant decrease in milling, appearance, as well as the cooking and taste quality of rice [7–9]. Several studies have shown that chalkiness is produced by a combination of internal genetic factors [5–8], such as maternal effects, endosperm effects, cytoplasmic effects, and external environmental factors [1,9]

(such as climatic conditions and cultivation methods). The genetic improvement of varieties is a time-consuming process [5,7], and the regulatory role of cultivation measures [2,10–15] has become increasingly prominent. Among cultivation measures, water and fertilizer are two factors that influence and interact with each other on the yield and quality of rice [4,10–13]. It has been shown that appropriate water management [14], reasonable nitrogen (N) fertilizer management [15,16], and water and N coupling [10] can significantly regulate the grain filling process and can affect grain plumpness, grain weight, and rice quality [10,14–16] through starch synthesis and accumulation in endosperm. Furthermore, water and fertilizer management can affect the microclimatic changes in canopy temperature, humidity, and light intensity [17–20] in the rice field by influencing the construction of the rice population canopy, such as leaf area index, plant height, and leaf spatial growth posture [11,12]. The canopy microclimate during the filling and setting period has a more direct effect on rice quality [10,17,18]. Previous studies by our group and other researchers have confirmed the coupling effect of water and N on rice yield formation and N utilization efficiency [4,11,13], which could simultaneously improve rice yield and nitrogen use efficiency. Suitable irrigation and N fertilizer application interactions would be beneficial for increasing the filling rate of inferior rice grains and grain filling after flowering [10] and promoting carbon and N metabolism [12], which in turn would improve yield and water–fertilizer utilization efficiency [11,13]. However, studies on the effects of water and N on the canopy microclimate of rice populations have focused on either of these factors of N fertilizer [16,20] or water [17], and there are only a few studies on whether there is an interaction effect between water and N on rice canopy microclimate, and on the relationship between canopy microclimate and chalkiness traits of superior and inferior grains during the grain filling stage under water–N interaction.

Therefore, in order to reduce rice chalkiness by optimized irrigation and nitrogen management, in this study, we investigated the effects of water and N interaction on grain filling, canopy microclimate, and chalkiness of grains, and explored the relationship between grain filling characteristics and canopy microclimate and grain chalkiness under water–N interaction, providing a theoretical and practical basis for reducing rice chalkiness and improving rice quality.

## 2. Materials and Methods

### 2.1. Test Materials and Experimental Design

Based on previous studies [10,12] that determined stable and efficient N application (150 kg·hm$^{-2}$), the modern agricultural research and development base of Sichuan Agricultural University, Chongzhou, Chengdu (103°38′ E, 30°33′ N; altitude 520.6 m) conducted a study in 2019 and 2020. The soil was sandy loam, with the particle composition of sand 45%, clay 35%, silt 20%, 1.25 g·cm$^{-3}$ bulk density, pH 6.41, 25.6 g·kg$^{-1}$ organic matter ($K_2Cr_2O_7$) (using the volumetric method), 1.71 g·kg$^{-1}$ total N (using the Kjeldahl method, FOSS-8400, Sweden), 25.2 mg·kg$^{-1}$ available phosphorus (using the Olsen method), and 154.7 mg·kg$^{-1}$ available potassium (using flame spectrometry after $NH_4OAc$ extraction) of composite topsoil samples (0–20 cm). The rice (Oryza sativa) variety 'F you 498 ' (late-maturing hybrid rice, growth period: 147–152 d) was tested. The seeds were sown on 22 April using a water direct seeder (Shanghai STAR Modern Agricultural Machinery Co., Ltd., Shanghai, China, Machine Model 2BDXS-10CP), and the rice grains were harvested on 3 September in both years. Chemical pesticides were used to avoid yield losses and experimental errors due to weeds, insects, and diseases. Each plot was 49.5 m$^2$, with a row spacing of 25 cm × 20 cm grown in triplicate for each treatment. The split-plot experiment was designed with the main plot (three irrigation modes) and the subplot (three N fertilizer application methods), as well as no N treatment. The irrigation modes were as follows:

(1)  Flooding irrigation (W$_1$): from the two-leaf stage of directly seeded rice, with the water layer continuously maintained at 1 to 2 cm, and natural drying was carried out 1 week before maturity;

(2) Dry–wet alternating irrigation ($W_2$): from the two-leaf stage of directly seeded rice, with irrigation at a soil water potential of $-25$ kPa and establishment of a 1 to 2 cm water layer, and measurement of soil water potential with a vacuum negative soil pressure meter (developed by the Institute of Soil Science, Chinese Academy of Sciences);

(3) Dry alternating irrigation ($W_3$): with irrigation at a soil water potential of $-40$ kPa, requiring a water-free surface layer.

The N fertilizer delivery patterns were as follows: 30%:50%:20% ($N_1$), 30%:30%:40% ($N_2$), and 30%:10%:60% ($N_3$) for base (1 d before sowing):tiller (20 d after sowing):panicle fertilizer (68 d after sowing), respectively, and no N treatment ($N_0$). Phosphate fertilizer (calcium superphosphate) was used as the base fertilizer at a rate of 75 kg hm$^{-2}$ $P_2O_5$. Potassium fertilizer (potassium chloride) was applied twice in accordance with the base:panicle fertilizer ratio of 50%:50% at a rate of 150 kg hm$^{-2}$ $K_2O$. Forty-centimeter ridges were built between adjacent plots and wrapped with plastic mulch to prevent water and fertilizer runoff between plots. The volume of irrigation water was recorded accurately, and the same irrigation mode was used to ensure that the same amount of irrigation water was applied each time. Other field management measures were carried out according to field production.

*2.2. Measurement Items and Methods*

2.2.1. Measurement and Parameters of Grain Filling

At the heading stage, panicles with relatively consistent flowering times and panicle size were selected and tagged, 300 panicles were tagged with a tag in each plot, and the tagged panicles were collected twice at 7 d intervals from flowering to maturity (within each 7 d period, the first sampling was on Day 3 and the second sampling was on Day 7). Ten panicles were collected each time, and the superior and inferior grains were removed. Following this, unfertilized empty grains were removed and other grains were dried, deshelled, and weighed. Superior and inferior grains were classified according to the following criteria: kernels attached to the top three primary peduncles of the panicle were considered to be superior grains, with the exception of the second top grain; and kernels attached to the three primary peduncles at the base of the panicle on the secondary peduncles were considered to be inferior grains, except for the top first grain. Richards' equation was used for fitting based on the method of Sun et al. [10] and Yang et al. [14].

$$W = A/(1 + Be^{-kt})^{1/N} \tag{1}$$

where $W$ is the rice grain weight (mg); $A$ is the final rice grain weight; $t$ is the time after flowering (d); and $B$, $k$, and $N$ are the equation parameters. $R^2$ is the coefficient of determination, indicating the fit of the equation. The grain filling rate G (mg·Grain$^{-1}$·d$^{-1}$) was obtained by taking the derivative of Equation (1):

$$G = ktBe^{-kt}/N(1 + Be^{-kt})^{(N+1)/N} \tag{2}$$

Calculation was performed based on the above parameters:

(1) $R_0$ (starting growth potential of grain) $= k/N$
(2) $W_{max}$ (grain weight at maximum filling rate) $= W = A(N + 1)^{-1/N}$
(3) $T_{max}$ (time of maximum growth rate) $= (\ln B - \ln N)/k$
(4) $GR_{mean}$ (mean filling rate) $= Ak/2(N + 2)$
(5) D (active growth period) $= 2(N + 2)/k$
(6) Samples were divided into early-, middle-, and late-filling stages, which are termed the $0$–$T_1$, $T_1$–$T_2$, and $T_2$–$T_{99}$ stages, respectively.

$$T_1 = -\ln[(N^2 + 3N + N \cdot \sqrt{N^2 + 6N + 5})/2B]/k$$

$$T_2 = -\ln[(N^2+3N - N\cdot\sqrt{N^2+6N+5})/2B]/k$$

$$T_{99}(\text{effective filling time}) = -\ln\{[(100/99)^N - 1]/B\}/k$$

Grain filling accumulation was calculated based on the three phases ($T_1$, $T_2$, and $T_{99}$) after flowering introduced in Equation (1). The mean grain filling rate (MGR) of the three filling periods before, during, and after the filling period was calculated based on the three phases and filling material accumulation, as well as the ratio of grain filling contributed to *A* value (RGC) in each phase.

### 2.2.2. Canopy Microclimate

The MX2301 thermometer and MX2305 light intensity meter (Onset Hobo Inc., MA, USA) were installed in each plot before heading at 1/2 the height of the panicle. The instruments were adjusted to record data every 10 min until rice harvest, and the following markers were calculated after reading and exporting data using Bluetooth on mobile phones.

(1) Total temperature difference between different filling stages = $\sum$ (highest temperature of the day–lowest temperature) for each filling stage.
(2) Average daily temperature difference at different filling stages = total temperature difference at each filling stage/number of days.
(3) Effective cumulative temperature at different filling stages = $\sum$ (average daily temperature of $-10$) for each filling stage.
(4) Average daily light intensity at different filling stages = total light intensity at each filling stage/number of days.

### 2.2.3. Chalky Grain Rate and Chalkiness Degree Measurements

A total of 100 panicles were harvested from each plot at the time of heading and labeled, before being threshed and stored in separate samples, according to the superior and inferior grains. The samples were naturally dried for 2 months, and, after screening and the removal of impurities, the chalky grain rate and chalkiness of superior and inferior grains were determined according to the methods described by Yoshioka et al. [21] and Chen et al. [22] with some modification. The samples of superior and inferior grain each at 50 g were taken from each plot. Rough rice was de-husked using a husker (JLGJ 4.5, Taizhou Cereal and Oil Instrument Co., Ltd., Zhejiang, China). From each sample, 20 g of brown rice was taken for milling using a Miller (JNM-III, China Grain Reserve Chengdu Research Institute Co., Ltd., Chengdu, China), and then head rice was divided from broken rice. A digital image was created by placing the head rice of each sample on the scanner, and the appearance quality indicators of chalky grain rate and chalkiness degree were analyzed by Image analyzing software (JMWT-12, Beijing Dongfu Jiuheng Instrument Co., Ltd., Beijing, China and Japan Satake Co., Ltd., Hiroshima, Japan)

### *2.3. Statistical Analysis*

Date analysis and graphing functions were performed using SAS 8.1 (SAS Institute, Cary, NC, USA) and SigmaPlot 12.0 (Systat Software Inc., Chicago, IL, USA), respectively. Means were tested by least significant difference (LSD) at the 0.05 level (LSD 0.05).

## 3. Results and Analysis

### *3.1. Grain Filling Characteristics*

#### 3.1.1. Dynamics of Increasing Grain Weight and Grain Filling Rate

The dynamic changes of superior and inferior grains weight after flowering in directly seeded rice were consistent with the Richards model, with coefficients of determination above 0.99 under different irrigation modes and N fertilizer regimes (Table 1, Figures 1 and 2). The trend of the effect of N fertilizer application on the final weight (*A*) of superior and inferior grains was the same under all irrigation modes; all were optimal in the $N_1$ treatment (Table 1), while increasing the proportion of panicle fertilizer in directly seeded

rice ($N_2$ and $N_3$ treatments) was not conducive to the maximum filling of the superior and inferior grains. With regards to each filling parameter, the $R_0$ of both the superior and inferior grains was highest in $N_0$ treatment under all irrigation modes, and the $R_0$ of both the superior and inferior grains decreased as the proportion of N panicle fertilizer increased under all irrigation modes, except for superior grains under $W_3$ treatment, which first increased and then decreased as the proportion of N panicle fertilizer was increased. The $T_{max}$ and the active growth period (D) delayed and increased, respectively, with increasing N panicle fertilizer percentage under each irrigation mode (Figures 1D–F and 2D–F). Moreover, $T_{max}$ presented as $W_1 > W_2 > W_3$ under the same N fertilizer treatment, indicating that water deficit irrigation caused the $T_{max}$ to occur earlier, while increasing the proportion of N panicle fertilizer led to delayed $T_{max}$, indirectly proving the existence of a significant mutual regulation effect between water and fertilizer. The grain weight ($W_{max}$) at maximum filling rate showed $W_1 > W_2 > W_3$. The $GR_{max}$ and $GR_{mean}$ of superior and inferior grains were optimal in $W_2$ treatment under all irrigation modes. Under the same irrigation mode, the $GR_{max}$ and $G_{mean}$ of the superior and inferior grains showed a decreasing trend with the increase in the N panicle fertilizer ratio. The N panicle fertilizer ratio higher than 40% or W3 treatment was not conducive to the grain filling of direct seeding rice.

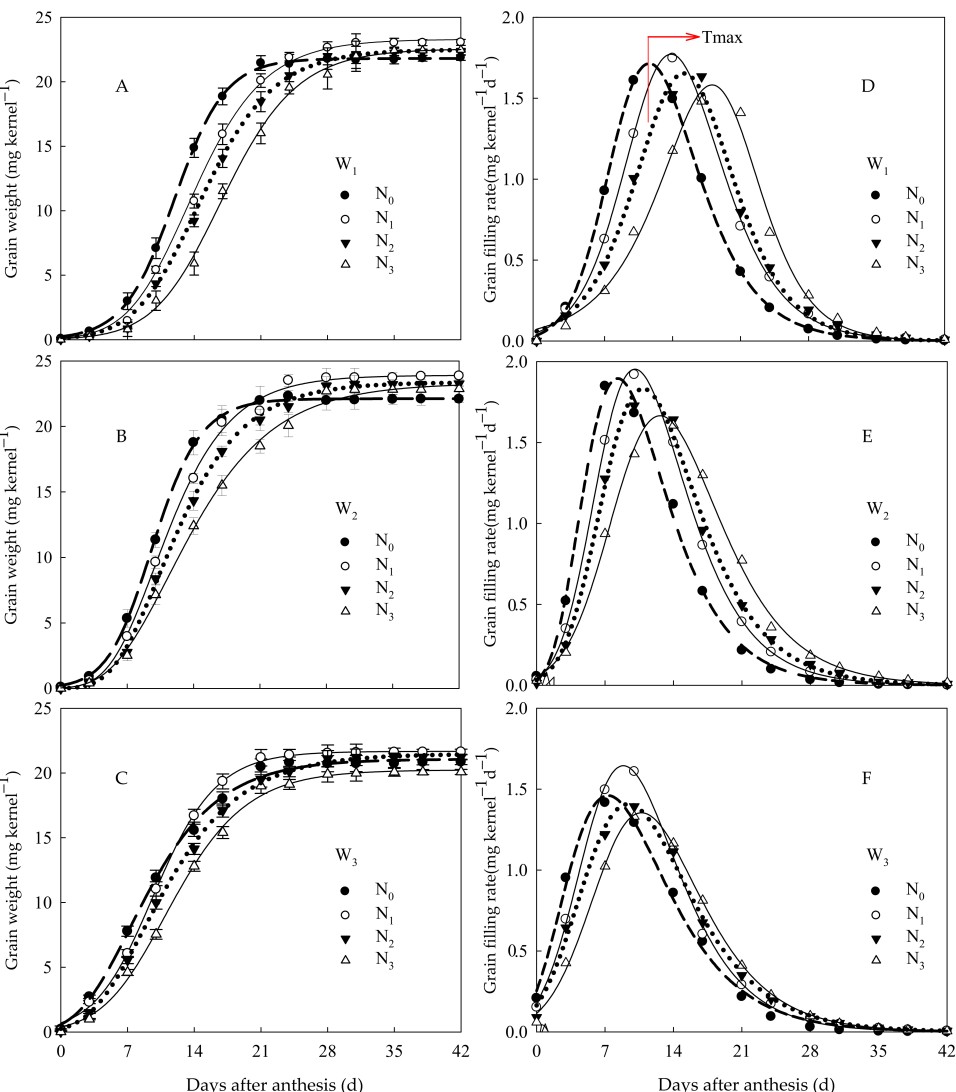

**Figure 1.** Grain filling process (**A–C**) and rate (**D–F**) of superior grains in directly seeded rice under water–nitrogen interaction (2019–2020).

**Table 1.** Parameters of the Richard equation and grain filling of directly seeded rice under water–nitrogen interaction.

| Irrigation Mode | Grain Position | N Application | Richard Equation Parameters | | | | | | | | Grain Filling Parameters | | |
|---|---|---|---|---|---|---|---|---|---|---|---|---|---|
| | | | $A$ | $B$ | $k$ | $N$ | $R^2$ | $R_0$ | $T_{max}$ (d) | D (d) | $W_{max}$ (mg·kernel$^{-1}$) | $GR_{max}$ (mg·kernel$^{-1}$·d$^{-1}$) | $GR_{mean}$ (mg·kernel$^{-1}$·d$^{-1}$) |
| $W_1$ | Superior grain | $N_0$ | 21.16 | 6.77 | 0.257 | 0.343 | 0.9963 | 0.750 | 11.61 | 18.23 | 8.96 | 1.71 | 1.16 |
| | | $N_1$ | 23.71 | 22.83 | 0.262 | 0.604 | 0.9963 | 0.434 | 13.85 | 19.86 | 10.84 | 1.77 | 1.19 |
| | | $N_2$ | 22.41 | 66.23 | 0.285 | 0.882 | 0.9976 | 0.323 | 15.17 | 20.25 | 10.94 | 1.65 | 1.11 |
| | | $N_3$ | 22.39 | 517.70 | 0.324 | 1.487 | 0.9983 | 0.218 | 18.07 | 21.53 | 12.13 | 1.58 | 1.04 |
| | Inferior grain | $N_0$ | 15.76 | 28.16 | 0.238 | 0.456 | 0.9971 | 0.523 | 17.30 | 20.61 | 6.91 | 1.13 | 0.76 |
| | | $N_1$ | 18.14 | 765.46 | 0.346 | 0.962 | 0.9979 | 0.360 | 19.29 | 17.11 | 9.00 | 1.59 | 1.06 |
| | | $N_2$ | 17.54 | 3612.99 | 0.349 | 1.418 | 0.9989 | 0.246 | 22.50 | 19.61 | 9.41 | 1.36 | 0.89 |
| | | $N_3$ | 17.30 | 4637.02 | 0.332 | 1.407 | 0.9973 | 0.236 | 24.41 | 20.53 | 9.27 | 1.28 | 0.84 |
| $W_2$ | Superior grain | $N_0$ | 21.85 | 0.12 | 0.236 | 0.020 | 0.9928 | 11.790 | 7.54 | 17.13 | 8.12 | 1.88 | 1.28 |
| | | $N_1$ | 24.81 | 1.71 | 0.236 | 0.159 | 0.9972 | 1.479 | 10.07 | 18.34 | 9.61 | 1.95 | 1.35 |
| | | $N_2$ | 24.15 | 1.72 | 0.222 | 0.154 | 0.9969 | 1.445 | 10.90 | 19.42 | 9.53 | 1.83 | 1.24 |
| | | $N_3$ | 24.01 | 4.03 | 0.208 | 0.290 | 0.9976 | 0.718 | 12.63 | 21.99 | 10.31 | 1.66 | 1.10 |
| | Inferior grain | $N_0$ | 17.25 | 8.14 | 0.223 | 0.224 | 0.9966 | 0.996 | 16.15 | 19.97 | 7.00 | 1.27 | 0.86 |
| | | $N_1$ | 21.30 | 26.38 | 0.237 | 0.330 | 0.9993 | 0.718 | 18.49 | 19.67 | 8.98 | 1.60 | 1.08 |
| | | $N_2$ | 18.78 | 107.82 | 0.263 | 0.583 | 0.9987 | 0.450 | 19.89 | 19.68 | 8.54 | 1.42 | 0.95 |
| | | $N_3$ | 18.69 | 331.23 | 0.263 | 0.828 | 0.9986 | 0.318 | 22.76 | 21.49 | 9.02 | 1.30 | 0.87 |
| $W_3$ | Superior grain | $N_0$ | 20.29 | 0.11 | 0.196 | 0.025 | 0.9943 | 7.780 | 7.31 | 20.71 | 7.56 | 1.44 | 0.98 |
| | | $N_1$ | 22.05 | 1.76 | 0.226 | 0.236 | 0.9950 | 0.958 | 8.89 | 19.77 | 8.99 | 1.64 | 1.12 |
| | | $N_2$ | 20.85 | 1.10 | 0.199 | 0.171 | 0.9970 | 1.164 | 9.39 | 21.84 | 8.28 | 1.41 | 0.95 |
| | | $N_3$ | 19.96 | 2.99 | 0.210 | 0.307 | 0.9958 | 0.684 | 10.84 | 21.97 | 8.36 | 1.34 | 0.91 |
| | Inferior grain | $N_0$ | 14.71 | 6.21 | 0.228 | 0.278 | 0.9966 | 0.820 | 13.62 | 19.98 | 6.09 | 1.09 | 0.74 |
| | | $N_1$ | 16.84 | 6.07 | 0.210 | 0.257 | 0.9983 | 0.817 | 15.06 | 21.50 | 6.92 | 1.16 | 0.78 |
| | | $N_2$ | 16.57 | 12.68 | 0.193 | 0.395 | 0.9978 | 0.489 | 17.97 | 24.82 | 7.13 | 0.99 | 0.67 |
| | | $N_3$ | 16.45 | 23.49 | 0.189 | 0.499 | 0.9993 | 0.379 | 20.38 | 26.44 | 7.31 | 0.92 | 0.62 |

$A$: final weight of a kernel; $B$: initial parameter; $k$: growth rate parameter; $N$: shape parameter; $R^2$: decisive factor; $R_0$: initial grain filling potential; $T_{max}$: the time reaching the maximum grain filling rate; $GR_{max}$: maximum grain filling rate; $W_{max}$: weight of a kernel at the time of maximum grain filling rate; $GR_{mean}$: mean grain filling rate; D: active grain filling period.

Vertical bars represent ±S.E. of the mean. The S.E. was calculated across three replicates for each year and average across 2 years. The experiment was conducted at Sichuan Agricultural University Farm, Chongzhou, Sichuan Province, southwest China (2019–2020).

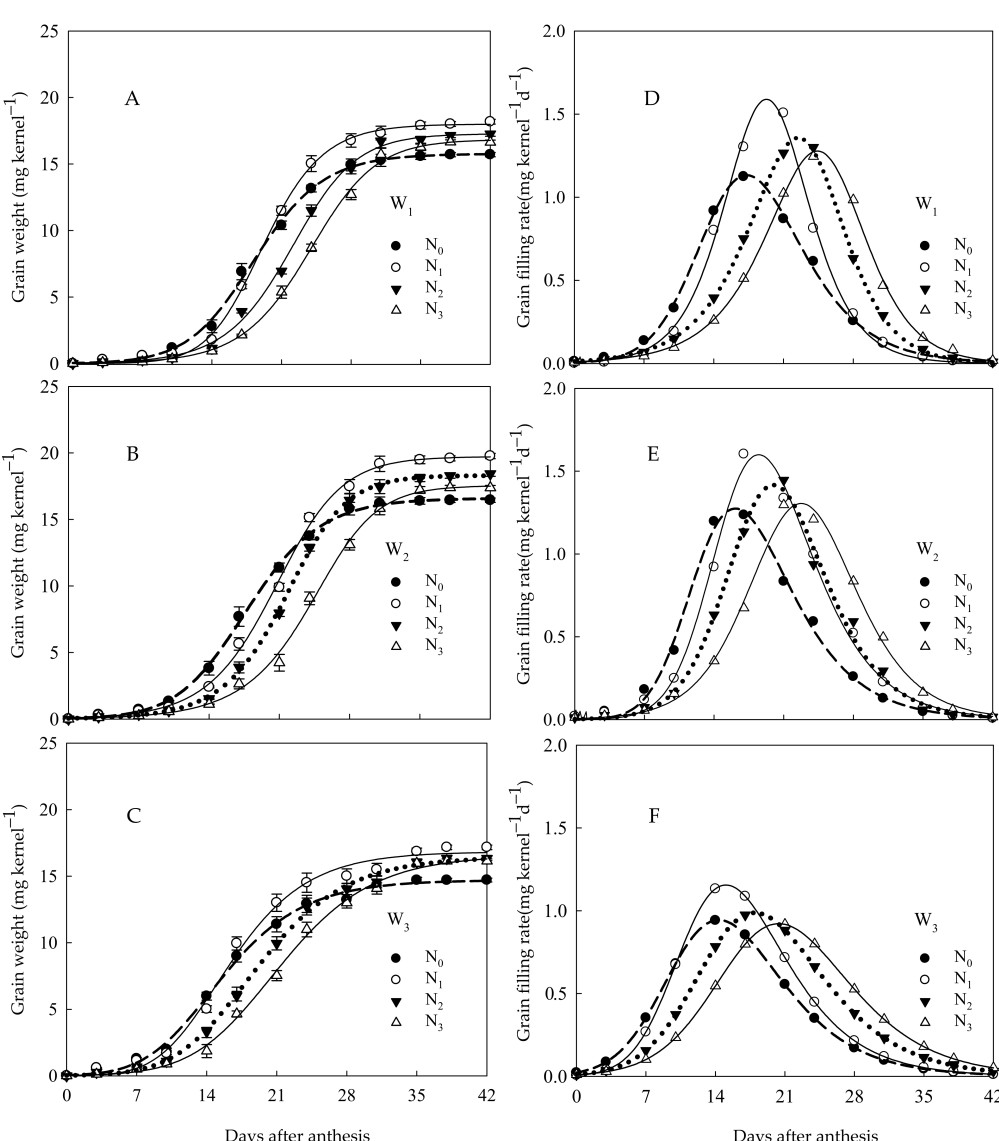

**Figure 2.** Grain-filling process (**A–C**) and rate (**D–F**) of inferior grains in directly seeded rice under water–nitrogen interaction (2019–2020).

Vertical bars represent ± S.E. of the mean. The S.E. was calculated across three replicates for each year and average across 2 years. The experiment was conducted at Sichuan Agricultural University Farm, Chongzhou, Sichuan Province, southwest China (2019–2020).

### 3.1.2. Early-, Middle-, and Late-Stage Grain Filling Characteristics

As shown in Table 2, the number of days of filling for both superior and inferior grains of directly seeded rice in the early-filling stage increased as the proportion of N panicle fertilizer increased under different water and N treatments. Except for the $N_0$ treatment, the mean grain filling rate (MGR) of superior grains in $W_1$ treatment increased as the proportion of N panicle fertilizer increased, while the inferior grains showed a decreasing trend as the proportion of N panicle fertilizer increased. The MGR of $N_1$ treatment was higher under $W_2$ treatment, while the MGR of inferior grains showed a decreasing trend as

the proportion of N panicle fertilizer increased. The MGR of inferior and superior grains in $W_3$ treatment decreased as the proportion of N panicle fertilizer increased. In addition, the ratio of the grain filling which contributed to the final grain weight (RGC) of the early-filling grains showed an increasing trend as the proportion of N panicle fertilizer increased under each irrigation mode. The RGC of both superior and inferior grains was above 55% in the mid-filling stage, although there was a small number of days of filling in this period. The MGR of superior and inferior grains showed an overall performance of $N_1 > N_2 > N_3$ in the late-filling stage. Moreover, with the increase in the N panicle fertilizer ratio, the MGR of superior and inferior grains showed a downward trend, and the RGC also decreased significantly.

### 3.2. Canopy Microclimate during Different Grain Filling Stages

Based on the number of days of each filling stage shown in Table 2, the average daily temperature difference, total temperature difference (Table 3), effective cumulative temperature, and average daily light intensity (Table 4) were calculated for each filling stage of superior and inferior grains under different water and N treatment. There were significant interaction effects of irrigation mode and N fertilizer application on the above four indicators of each filling stage for both superior and inferior grains in directly seeded rice. As shown in Table 3, the average daily temperature difference between superior and inferior grains in $W_1$ treatment was lower than that in the $W_2$ and $W_3$ treatments in each filling stages. However, the total temperature difference between superior and inferior grains in the early-filling stage was $W_1 > W_2 > W_3$, and the total temperature difference in $W_1$ treatment was lower than that in $W_2$ and $W_3$ treatments in middle- and late-filling stages. Under each irrigation mode, the average daily temperature difference between superior and inferior grains of directly seeded rice at each filling stage showed different decreasing trends as the proportion of N panicle fertilizer increased. In particular, a proportion of N panicle fertilizer of 60% would lead to a significant decrease in the average daily temperature difference and the total temperature difference in the canopy. In addition, as seen in Table 4, the total temperature difference in the early-filling stage for both superior and inferior grains presented as $W_1 > W_2 > W_3$, and the total temperature difference in the late-filling stage was lower in the $W_1$ treatment than in the $W_2$ and $W_3$ treatments. Under each irrigation mode, the main difference in effective cumulative temperature between superior and inferior grains in 20% and 60% N panicle fertilizer rate was in the early-filling stage, and the effective cumulative temperature of superior and inferior grains increased significantly as N panicle fertilizer rate increased in early-filling stage. The effects of irrigation modes and N fertilizer application on the average daily light intensity of superior and inferior grains of directly seeded rice at each filling stage were consistent with the average daily temperature difference.

**Table 2.** Grain filling characteristics at the early-, middle-, and late-filling stages of directly seeded rice under water–nitrogen interaction.

| Irrigation Mode | Grain Position | N Application | Early-Filling Stage | | | Middle-Filling Stage | | | Late-Filling Stage | | |
|---|---|---|---|---|---|---|---|---|---|---|---|
| | | | Days (d) | MGR (mg·kernel$^{-1}$·d$^{-1}$) | RGC % | Days (d) | MGR (mg·kernel$^{-1}$·d$^{-1}$) | RGC % | Days (d) | MGR (mg·kernel$^{-1}$·d$^{-1}$) | RGC % |
| $W_1$ | Superior grain | $N_0$ | 7.32 | 0.365 | 12.64 | 8.58 | 1.490 | 60.37 | 13.60 | 0.404 | 25.99 |
| | | $N_1$ | 9.30 | 0.415 | 16.28 | 9.11 | 1.547 | 59.44 | 12.98 | 0.425 | 23.28 |
| | | $N_2$ | 10.67 | 0.415 | 19.76 | 9.01 | 1.448 | 58.26 | 11.64 | 0.403 | 20.98 |
| | | $N_3$ | 13.60 | 0.432 | 26.21 | 8.94 | 1.392 | 55.52 | 9.71 | 0.398 | 17.27 |
| | Inferior grain | $N_0$ | 12.50 | 0.180 | 14.26 | 9.60 | 0.985 | 60.00 | 14.50 | 0.269 | 24.74 |
| | | $N_1$ | 15.52 | 0.244 | 20.69 | 7.54 | 1.393 | 57.91 | 9.50 | 0.389 | 20.40 |
| | | $N_2$ | 18.40 | 0.242 | 25.33 | 8.20 | 1.194 | 55.88 | 9.07 | 0.341 | 17.67 |
| | | $N_3$ | 20.11 | 0.219 | 25.45 | 8.60 | 1.124 | 55.89 | 9.54 | 0.321 | 17.66 |
| $W_2$ | Superior grain | $N_0$ | 3.42 | 0.486 | 7.62 | 8.24 | 1.617 | 60.96 | 15.39 | 0.432 | 30.42 |
| | | $N_1$ | 5.70 | 0.421 | 9.77 | 8.75 | 1.690 | 60.87 | 15.15 | 0.454 | 28.36 |
| | | $N_2$ | 6.26 | 0.377 | 9.86 | 9.27 | 1.584 | 60.79 | 16.10 | 0.426 | 28.35 |
| | | $N_3$ | 7.43 | 0.396 | 11.86 | 10.39 | 1.445 | 60.52 | 16.88 | 0.391 | 26.62 |
| | Inferior grain | $N_0$ | 11.40 | 0.164 | 10.85 | 9.49 | 1.104 | 60.69 | 15.91 | 0.298 | 27.46 |
| | | $N_1$ | 13.86 | 0.191 | 12.45 | 9.26 | 1.389 | 60.41 | 14.78 | 0.377 | 26.14 |
| | | $N_2$ | 15.46 | 0.192 | 15.84 | 9.01 | 1.235 | 59.57 | 13.01 | 0.339 | 23.60 |
| | | $N_3$ | 17.95 | 0.200 | 19.12 | 9.62 | 1.142 | 58.50 | 12.65 | 0.317 | 21.39 |
| $W_3$ | Superior grain | $N_0$ | 2.34 | 0.670 | 7.71 | 9.95 | 1.243 | 60.96 | 18.54 | 0.332 | 30.34 |
| | | $N_1$ | 4.18 | 0.580 | 10.02 | 9.38 | 1.426 | 60.81 | 15.64 | 0.385 | 28.17 |
| | | $N_2$ | 4.20 | 0.501 | 10.04 | 10.41 | 1.217 | 60.80 | 17.93 | 0.328 | 28.16 |
| | | $N_3$ | 5.61 | 0.431 | 12.11 | 10.27 | 1.175 | 60.48 | 16.56 | 0.318 | 26.41 |
| | Inferior grain | $N_0$ | 8.92 | 0.207 | 12.57 | 10.84 | 0.819 | 60.39 | 17.23 | 0.222 | 26.05 |
| | | $N_1$ | 9.96 | 0.192 | 11.36 | 10.19 | 1.002 | 60.61 | 16.81 | 0.271 | 27.03 |
| | | $N_2$ | 12.16 | 0.182 | 13.40 | 11.62 | 0.858 | 60.21 | 18.01 | 0.234 | 25.40 |
| | | $N_3$ | 14.25 | 0.172 | 14.86 | 12.26 | 0.803 | 59.84 | 18.20 | 0.220 | 24.30 |

MGR: mean grain filling rate; RGC: ratio of the grain filling which contributed to the final grain weight.

**Table 3.** Effects of irrigation mode and N application on canopy temperature difference (°C) at early-, middle-, and late-grain filling stages of directly seeded rice.

| Irrigation Mode | Grain Position | N Application | Daily Average Temperature Difference | | | Total Temperature Difference | | |
|---|---|---|---|---|---|---|---|---|
| | | | Early Stage | Middle Stage | Late Stage | Early Stage | Middle Stage | Late Stage |
| $W_1$ | Superior grain | $N_0$ | 11.43 a | 14.57 a | 10.49 a | 83.67 c | 125.01 a | 142.66 a |
| | | $N_1$ | 10.63 b | 13.85 b | 10.25 ab | 98.81 b | 126.17 a | 133.05 a |
| | | $N_2$ | 10.50 b | 13.62 b | 10.01 b | 111.98 a | 122.72 a | 116.52 b |
| | | $N_3$ | 7.41 c | 10.00 c | 7.40 c | 100.78 b | 89.40 b | 71.85 c |
| | Inferior grain | $N_0$ | 12.25 a | 12.99 a | 10.12 a | 153.13 c | 124.66 a | 146.70 a |
| | | $N_1$ | 11.79 a | 12.50 ab | 9.84 ab | 182.98 b | 94.23 b | 93.52 b |
| | | $N_2$ | 10.76 b | 12.44 b | 9.68 b | 197.98 a | 102.05 ab | 87.77 b |
| | | $N_3$ | 7.91 c | 9.81 c | 7.83 c | 159.07 c | 84.33 c | 74.68 c |
| $W_2$ | Superior grain | $N_0$ | 11.72 a | 14.19 a | 11.93 a | 40.07 c | 116.93 b | 183.60 a |
| | | $N_1$ | 11.01 ab | 14.18 a | 11.15 b | 62.73 ab | 124.08 ab | 168.92 b |
| | | $N_2$ | 10.89 b | 14.07 a | 10.90 b | 68.17 a | 130.43 a | 175.49 ab |
| | | $N_3$ | 7.74 c | 9.70 c | 7.45 c | 57.51 b | 100.78 c | 125.76 c |
| | Inferior grain | $N_0$ | 12.95 a | 13.73 a | 11.04 a | 147.63 c | 130.27 a | 175.62 a |
| | | $N_1$ | 12.24 ab | 13.44 a | 11.01 a | 169.65 b | 124.45 a | 162.69 b |
| | | $N_2$ | 11.58 b | 11.27 b | 9.63 b | 179.03 a | 101.59 b | 125.25 c |
| | | $N_3$ | 8.39 c | 10.45 c | 8.17 c | 150.60 c | 100.56 b | 103.37 d |
| $W_3$ | Superior grain | $N_0$ | 11.78 a | 13.33 a | 11.18 a | 27.55 b | 132.63 ab | 207.28 a |
| | | $N_1$ | 10.63 b | 13.16 ab | 11.07 ab | 44.41 a | 123.44 b | 173.13 c |
| | | $N_2$ | 10.31 b | 12.78 b | 10.71 b | 43.28 a | 133.04 a | 192.03 b |
| | | $N_3$ | 8.56 c | 11.14 c | 8.50 c | 43.00 a | 114.41 c | 140.76 d |
| | Inferior grain | $N_0$ | 13.28 a | 14.08 a | 11.38 a | 118.46 c | 152.59 a | 196.05 a |
| | | $N_1$ | 12.48 b | 13.93 a | 11.35 a | 124.30 c | 141.93 b | 190.87 ab |
| | | $N_2$ | 12.01 b | 13.18 b | 10.07 b | 146.04 a | 153.19 a | 181.30 b |
| | | $N_3$ | 9.55 c | 11.83 c | 9.27 c | 136.09 b | 145.04 b | 168.72 c |

Different letters in the same column under each irrigation mode and the same grain position meant significantly different among N treatment at $p < 0.05$.

Table 4. Effects of irrigation mode and N application on canopy effective accumulated temperature (°C) and daily average light intensity (lux) at the early-, middle-, and late-grain filling stages of directly seeded rice.

| Irrigation Mode | Grain Position | N Application | Effective Accumulated Temperature | | | Daily Average Light Intensity | | |
|---|---|---|---|---|---|---|---|---|
| | | | Early Stage | Middle Stage | Late Stage | Early Stage | Middle Stage | Late Stage |
| $W_1$ | Superior grain | $N_0$ | 104.99 d | 144.21 a | 224.31 a | 1212.13 a | 1441.44 a | 1031.46 a |
| | | $N_1$ | 136.76 c | 146.64 a | 207.69 b | 972.71 b | 1284.93 b | 965.14 b |
| | | $N_2$ | 167.10 b | 150.35 a | 191.31 c | 907.79 c | 1282.56 b | 867.01 c |
| | | $N_3$ | 217.78 a | 146.96 a | 158.88 d | 895.85 c | 909.64 c | 685.90 d |
| | Inferiorgrain | $N_0$ | 182.34 d | 223.94 a | 195.13 a | 1208.61 a | 1215.85 a | 912.53 a |
| | | $N_1$ | 250.13 c | 126.90 c | 159.23 b | 1086.11 b | 1000.47 b | 707.17 b |
| | | $N_2$ | 284.25 b | 124.52 c | 162.13 b | 1043.21 c | 901.97 c | 691.84 b |
| | | $N_3$ | 317.83 a | 142.16 b | 123.27 c | 939.18 d | 544.22 d | 488.49 c |
| $W_2$ | Superior grain | $N_0$ | 41.38 c | 125.19 c | 242.04 b | 1029.03 a | 1085.94 c | 1152.70 a |
| | | $N_1$ | 88.58 b | 142.77 b | 242.49 b | 996.67 b | 1289.00 a | 1159.25 a |
| | | $N_2$ | 88.74 b | 143.66 b | 256.68 a | 759.84 c | 1190.84 b | 874.16 b |
| | | $N_3$ | 103.91 a | 156.40 a | 259.93 a | 650.06 d | 655.17 d | 588.26 c |
| | Inferiorgrain | $N_0$ | 166.57 d | 123.18 c | 238.15 a | 1138.10 a | 1269.45 a | 891.78 a |
| | | $N_1$ | 215.18 c | 147.07 ab | 215.45 b | 1070.42 b | 1190.19 b | 822.70 b |
| | | $N_2$ | 232.24 b | 144.24 b | 191.06 c | 1018.44 c | 981.21 c | 810.69 b |
| | | $N_3$ | 276.24 a | 154.51 a | 165.32 d | 650.03 d | 629.20 d | 551.90 c |
| $W_3$ | Superior grain | $N_0$ | 31.17 c | 162.76 a | 314.96 a | 1135.55 a | 1228.73 a | 1185.28 a |
| | | $N_1$ | 61.58 b | 143.48 b | 263.08 c | 1179.10 a | 1071.52 b | 921.13 b |
| | | $N_2$ | 62.14 b | 161.55 a | 297.72 b | 866.85 b | 576.60 c | 830.90 c |
| | | $N_3$ | 92.79 a | 154.42 a | 247.40 d | 545.06 c | 504.54 d | 347.96 d |
| | Inferior grain | $N_0$ | 142.77 d | 190.96 a | 249.52 a | 1185.40 a | 1409.65 a | 980.15 a |
| | | $N_1$ | 155.66 c | 170.16 b | 255.48 a | 1028.27 b | 1359.72 b | 892.50 b |
| | | $N_2$ | 189.28 b | 196.36 a | 257.74 a | 689.28 c | 955.63 c | 711.18 c |
| | | $N_3$ | 214.05 a | 189.53 a | 220.26 b | 534.54 d | 631.66 d | 496.56 d |

Different letters in the same column under each irrigation mode and the same grain position meant significantly different among N treatment at *p* < 0.05.

*3.3. Grain Chalkiness*

　　The irrigation mode and N fertilizer application had significant or extremely significant reciprocal effects on chalky grain rate and chalkiness of superior and inferior grains of directly seeded rice, the effect of N fertilizer application on each marker was more significant than that of the irrigation treatment, and the trends were consistent in the 2 year study (Table 5). In terms of the effects of water and N treatments on chalky grain rate and chalkiness of superior grains, the effects of the same irrigation mode on rice chalkiness were not consistent; under the $W_1$ treatment, both the chalky grain rate and chalkiness showed a trend of initially decreasing and then significantly increasing as the proportion of N panicle fertilizer increased, of which a proportion of N panicle fertilizer of 40% yielded the best results. Under the $W_2$ and $W_3$ treatments, both the chalky grain rate and chalkiness degree showed different degrees of deterioration with the increase in nitrogen fertilizer ratio, and the best treatment was 20% of N panicle fertilizer. In terms of the effect of water and N treatments on the chalky grain rate and chalkiness of the inferior grains, N fertilizer application significantly reduced the chalky grain rate and chalkiness of the inferior grains under each irrigation mode compared with $N_0$. Under $W_1$ treatment, the chalky grain rate and chalkiness of inferior grains with 40% N panicle fertilizer ratio were the lowest. Under $W_2$ and $W_3$ treatments, the effects of N fertilizer application on the chalky grain rate and chalkiness were consistent with those of superior grains, and both were optimal at 20% N panicle fertilizer.

*3.4. Relationship of Grain Filling Parameters and Canopy Microclimate with Chalkiness under Water–N Interaction*

　　The correlation coefficients between the filling parameters, chalky grain rate, and chalkiness of superior and inferior grains in directly seeded rice under irrigation modes and N fertilizer applications were different (Table 6). The chalky grain rate and chalkiness of superior grains were negatively and significantly correlated with the $GR_{max}$ and $GR_{mean}$ ($r = -0.745$ **$\sim -0.773$ **), but were positively correlated with the active growth period (*D*). Compared with superior grains, the chalky grain rate and chalkiness of the inferior grains were more affected by filling parameters. Except for the insignificant correlation between the $T_{max}$ and chalkiness, the chalky grain rate and chalkiness of the inferior grains were significantly or extremely significantly negatively correlated with $T_{max}$, $GR_{max}$, $GR_{mean}$, and $W_{max}$ (r = $-0.506$ *$\sim -0.765$ **).

　　Furthermore, the relationship between the MGR of superior and inferior grain and the canopy microclimate indexes were closely related to chalky grain rate and chalkiness degree at the early-, middle-, and late-filling stages (Table 6). Compared with the early- and late-filling stages, the MGR of superior and inferior grains at the mid-filling stage was significantly negatively correlated with the chalkiness traits of superior and inferior grains (r = $-0.676$ *$\sim -0.750$ **). The highest correlation coefficient was observed at the mid-filling stage. In addition, the effect of MGR of superior grains on chalkiness was significantly higher than that of inferior grains at the mid-filling stage. The effects of mean daily temperature difference and mean daily light intensity on the chalky traits of superior grains showed the highest correlation coefficient (r = $-0.644$ *$\sim -0.718$ **) at the mid-filling stage, but their effects were less prominent than the effects of MGR of superior grain on chalkiness. On the contrary, the effects of mean daily temperature difference and mean daily light intensity on the chalkiness of inferior grains were mainly significant (r = $-0.571$ *$\sim -0.675$ *) at the early-filling stage, which was significantly greater than the MGR of inferior grain on grain chalkiness.

Table 5. Chalkiness characteristics of superior and inferior grain in directly seeded rice under water–nitrogen interaction.

| Irrigation Mode | N Application | Superior Grain | | | | Inferior Grain | | | |
|---|---|---|---|---|---|---|---|---|---|
| | | 2019 y | | 2020 y | | 2019 y | | 2020 y | |
| | | Chalky Grain Rate (%) | Chalkiness Degree (%) | Chalky Grain Rate(%) | Chalkiness Degree (%) | Chalky Grain Rate (%) | Chalkiness Degree (%) | Chalky Grain Rate (%) | Chalkiness Degree (%) |
| $W_1$ | $N_0$ | 21.51 b | 8.02 b | 20.07 b | 7.70 b | 44.69 a | 16.98 a | 47.88 a | 18.24 a |
| | $N_1$ | 18.49 c | 7.46 bc | 19.35 bc | 6.68 c | 33.97 c | 13.70 c | 35.56 bc | 12.27 c |
| | $N_2$ | 17.52 c | 6.95 c | 17.81 c | 5.53 d | 32.19 c | 12.77 d | 32.73 c | 10.15 d |
| | $N_3$ | 24.32 a | 9.14 a | 26.06 a | 9.83 a | 39.52 b | 14.74 b | 36.89 b | 14.14 b |
| $W_2$ | $N_0$ | 22.11 a | 8.57 a | 21.00 b | 7.05 b | 43.42 a | 16.44 a | 45.57 a | 16.38 a |
| | $N_1$ | 16.67 c | 5.91 c | 15.35 c | 5.19 c | 31.51 c | 11.49 c | 29.01 c | 10.08 c |
| | $N_2$ | 18.96 b | 7.63 b | 15.63 c | 5.61 c | 35.84 b | 14.42 b | 29.54 c | 10.61 c |
| | $N_3$ | 22.97 a | 8.90 a | 24.11 a | 8.87 a | 36.11 b | 14.69 b | 34.02 b | 11.56 b |
| $W_3$ | $N_0$ | 23.22 b | 9.73 ab | 23.25 b | 9.08 a | 46.82 a | 17.17 a | 52.89 a | 18.83 a |
| | $N_1$ | 20.84 c | 6.81 c | 18.75 c | 6.10 c | 36.11 c | 11.80 c | 32.48 c | 10.57 d |
| | $N_2$ | 21.93 bc | 8.87 b | 22.10 b | 7.96 b | 37.99 bc | 15.36 b | 38.29 b | 13.79 c |
| | $N_3$ | 27.02 a | 9.91 a | 30.53 a | 10.87 a | 40.24 b | 16.85 a | 40.29 b | 15.73 b |
| *F* Value | W | 46.95 ** | 151.43 ** | 88.29 ** | 221.60 ** | 17.55 ** | 17.38 ** | 52.78 ** | 116.85 ** |
| | N | 94.61 ** | 428.64 ** | 207.02 ** | 540.10 ** | 92.58 ** | 168.84 ** | 212.60 ** | 406.14 ** |
| | W×N | 3.90 ** | 44.89 ** | 9.64 ** | 49.90 ** | 3.32 * | 18.63 ** | 5.13 ** | 26.77 ** |

Different letters in the same column under each irrigation mode meant significant differences among N treatment at $p < 0.05$. W, irrigation mode; N, N application; W × N, irrigation mode and N application interaction. *, ** Significantly different at the 0.05 and 0.01 probability levels, respectively.

**Table 6.** Correlation coefficients between grain filling characteristics, canopy microclimate during the grain filling stage, and chalky characteristics of directly seeded rice under water–nitrogen interaction.

| | Index | Superior Grain | | Inferior Grain | |
|---|---|---|---|---|---|
| | | Chalky grain Rate (%) | Chalkiness Degree (%) | Chalky grain Rate (%) | Chalkiness Degree (%) |
| | $T_{max}$ | 0.058 | 0.049 | −0.506 * | −0.343 |
| | $GR_{max}$ | −0.745 ** | −0.747 ** | −0.604 * | −0.629 * |
| | $W_{max}$ | −0.175 | −0.194 | −0.765 ** | −0.676 * |
| | $D$ | 0.604 * | 0.613 * | 0.151 | 0.269 |
| | $GR_{mean}$ | −0.767 ** | −0.773 ** | −0.586 * | −0.614 * |
| Early stage | MGR | 0.181 | 0.200 | −0.359 | −0.404 |
| | Daily average temperature difference | −0.570 * | −0.507 * | −0.675 * | −0.581 * |
| | Total temperature difference | −0.300 | −0.296 | −0.616 * | −0.530 * |
| | Effective accumulated temperature | 0.031 | 0.009 | −0.537 * | −0.412 |
| | Daily average light intensity | −0.420 | −0.390 | −0.643 * | −0.571 * |
| Middle stage | MGR | −0.745 ** | −0.750 ** | −0.685 * | −0.676 * |
| | Daily average temperature difference | −0.717 ** | −0.671 * | −0.530 * | −0.503 |
| | Total temperature difference | −0.509 * | −0.430 | 0.375 | 0.408 |
| | Effective accumulated temperature | 0.356 | 0.452 | 0.443 | 0.464 |
| | Daily average light intensity | −0.718 ** | −0.644 * | −0.639 * | −0.523 * |
| Late stage | MGR | −0.718 ** | −0.727 ** | −0.677 * | −0.675 * |
| | Daily average temperature difference | −0.599 * | −0.565 * | −0.607 * | −0.574 * |
| | Total temperature difference | −0.228 | −0.167 | 0.436 | 0.342 |
| | Effective accumulated temperature | 0.070 | 0.142 | 0.339 | 0.239 |
| | Daily average light intensity | −0.651 * | −0.550 * | −0.557 * | −0.545 * |

$T_{max}$: the time reaching the maximum grain filling rate; $GR_{max}$: maximum grain filling rate; $W_{max}$: weight of a kernel at the time of maximum grain filling rate; $D$: active grain filling period; $GR_{mean}$: mean grain filling rate. *, ** Significantly different at the 0.05 and 0.01 robability levels, respectively.

## 4. Discussion

### 4.1. Relationship between Grain Filling Characteristics and Rice Chalkiness Traits in Directly Seeded Rice under Water–N Interaction

Unlike traditional manual transplanted and machine-planted rice, the vegetative growth period of directly seeded rice is significantly shorter [1,3], and the improvement in yield and rice quality is mainly determined by photosynthetic material production capacity and grain filling rate [4,9,10,12]. Numerous studies have shown that the genetic characteristics of rice varieties are significantly correlated with grain filling characteristics [14,16,23]. The rice variety Y Liangyou 2 with less amylose content might result from its higher $G_{max}$ and $G_{mean}$ of the inferior grain than that of Liangyoupei 9 [16]. Moreover, the differences in chalkiness traits are mainly determined by differences in genetic background and key enzymes for starch synthesis in the grain during the filling period [15,24], which are caused by the insufficient accumulation of starch and protein particles in endosperm. Both water and fertilizer cultivation management measures can significantly regulate the process of grain filling and rice quality, and are important tools to further improve the rice quality of breakthrough rice varieties [16,22,25–29]. Currently, there are several studies on water and fertilizer regulation of grain filling and chalkiness, but these mostly focused on single factors of N fertilizer [2,15,16] and water [14,29]. Previous studies on the role of water on grain filling regulation and chalkiness primarily dealt with water stress and the physiological and biochemical factors [9,14]. During the filling stage, dry–wet alternating irrigation with a soil water potential of $0 \sim -25$ kPa can promote the movement of stored material to the panicle and increase grain sucrose synthase, adenosine diphosphoglucose, pyrophosphorylase (ADPGPase), and starch synthase (SSase) activities, which facilitated grain filling and led to water saving and quality improvement [4,7,14]. In the present study, we found that the $T_{max}$ of superior and inferior grains was moderately brought forward, and the $GR_{mean}$ and $A$ increased under mild water stress in flood irrigation treatment, while the quality, $R_0$, $W_{max}$, and $GR_{max}$ should be further optimized (Table 1). In particular, this can increase the filling rate at the mid-filling stage (Table 2), and thus reduce grain chalkiness, which further enriched and complemented the results of previous studies [2,14–16,18]. Under severe water stress ($W_3$ treatment), the $T_{max}$ of superior and inferior grains was significantly brought forward, resulting in a longer $D$ of inferior grains and a significant reduction in the $GR_{mean}$ and $A$ (Table 1). This may be due to the fact that severe water stress led to a reduction in the grain filling rate, resulting in a rapid reduction in the water content of the grains, causing premature water loss and early maturation of the grains, which affected starch synthesis and accumulation [26,27], and eventually manifested as a significant increase in chalky grain rate and chalkiness (Table 5). Related to water management, there is a huge debate on the effect of the N fertilizer application period on rice chalkiness. This is particularly true for conclusions on the effects of addition of N fertilizer at the late stage on chalk formation [2,15,16,20]. Some studies have shown that chalky grain rate and chalkiness decreased as N fertilizer application was carried out earlier within a certain range of N application [15]. Other studies have shown that the chalky grain rate significantly reduced when N fertilizer application was carried out at an early stage [2,16]. The present study showed that the optimum N fertilizer application was less consistent under different irrigation modes; flooding irrigation combined with N fertilizer application in the ratio of 3:3:4 for base: tiller: panicle fertilizer was appropriate, while both dry–wet alternating and dry alternating irrigation with N fertilizer application in the ratio of 3:5:2 for base: tiller: panicle fertilizer was appropriate to reduce grain chalkiness. An increase in the N panicle fertilizer ratio up to 60% resulted in increased chalkiness, which could lead to unreasonable grain filling, prolonged $D$, decreased $GR_{max}$ and $GR_{mean}$ (Table 1), and uneven assimilated substance distribution, resulting in late grain maturation, which in turn increased chalkiness [4,30,31]. In summary, with the increase in irrigation water stress, the key parameters ($T_{max}$, $GR_{max}$, and $W_{max}$) of both superior and inferior grains were significantly advanced under the same N fertilizer application. However, the key parameters ($T_{max}$, $D$, $GR_{max}$, and $W_{max}$) of both superior and inferior grains under the

same irrigation pattern were each significantly delayed or reduced with the increase in N panicle fertilizer ratio (Figure 1D–F and Figure 2D–F). It is indirectly proven that there are equilibrium points and obvious water–N interaction between water management and N fertilizer application, which may also be an important pathway of water–N coupling to regulate grain filling.

The relationship between grain filling characteristics and rice chalkiness in directly seeded rice under water–N interaction has received little attention. Previous studies only compared different varieties and found that chalkiness and grain filling dynamic parameters generally showed an upwards parabolic relationship [15,16]; however, varieties with relatively smooth filling rates were less chalky, and treatments or varieties with more fluctuating filling rates were chalkier [16]. It was initially determined that rice grain filling had some relationship with chalk formation [26,29,32]. In this study, we found that water–N coupling influenced chalkiness traits by regulating the process of grain filling, but different pathways were regulated in different grain positions, which may be explained by the preferential input of assimilated substances in superior grains and inhibitory effects on filling in the inferior grains [10,30,31,33,34]. Under water–N interaction, the chalky grain rate and chalkiness of superior grains were extremely significantly correlated with the parameters $GR_{max}$, $GR_{mean}$, and $D$ of the filling curve during the filling stage, indicating that the formation of chalkiness in superior grains could be reduced by decreasing the $D$ of superior grains and increasing $GR_{max}$ and $GR_{mean}$. In addition, relative to the superior grains, the chalky grain rate and chalkiness of the inferior grains are affected by water–N interaction and the filling parameters of the inferior grains, and chalkiness was significantly or extremely significantly correlated with $T_{max}$, $GR_{max}$, $GR_{mean}$, and $W_{max}$ through the chalky grain rate and chalkiness of the inferior grains, suggesting that increasing $GR_{max}$, $GR_{mean}$, and $W_{max}$ through the earlier appearance of $T_{max}$ of the inferior grains can decrease chalk formation in inferior grains. The present study only analyzed the relationship between grain filling characteristics of different grain positions and the chalky grain rate and chalkiness degree from water and N treatments. However, the basis and regulatory pathways of how water–N interactions affect grain filling and chalk formation from gene expression, physiological and biochemical, and hormone levels are yet to be further studied.

*4.2. Relationship between Canopy Microclimate and Rice Chalkiness Traits at Fruiting Stage under Water–N Interaction*

It is difficult for humans to change the regional climatic macro-environment in which crops grow, and canopy microclimate is the most direct micro-environmental factor affecting crop growth and development [17–20]. Studies have shown that the canopy microclimate of rice panicles at the fruiting stage is significantly different from the meteorological factors, such as temperature, fitness, and light in the upper 1 m of the canopy [17], while the canopy microclimate can be artificially regulated through the selection of different plant varieties and cultivation measures [9,18,21]. There have been many studies on the effects of field environment on rice chalkiness traits, mainly focusing on the effect of temperature on rice quality [24,26,27,35]. These studies found that a higher nighttime temperature increases chalk formation [27,28]; rice chalkiness is positively correlated with temperature at maturity, especially in the second week after heading when the average daily temperature is most closely correlated with chalkiness [22,24]; and the suitable climatic conditions under which rice sets are changed mainly through measures such as regulating the sowing period [22]. Fewer studies have been conducted on the effects of canopy microclimate characteristics on rice chalkiness traits during the seeding stage.

The findings of the present study showed that there was a significant mutual regulatory effect of water and N treatments on canopy microclimate (Tables 3 and 4). There were significant differences in the canopy microclimate at the early-, middle-, and late-filling stages of superior and inferior grains under different water and N treatments. The effect of N fertilizer on canopy microclimate was significantly higher than that of irrigation mode. With the proportion of delayed N fertilizer application increased, the average daily

temperature difference and average daily light intensity showed a decreasing trend in different degrees. The possible reasons for the increase in rice chalkiness caused by an increase in the percentage of postponing N topdressing are the increase in leaf area index, closure, and humidity of the canopy [12,14]; the decrease in light transmission [12,26]; and the night temperature not decreasing quickly [27,28], resulting in an increase in the average daily temperature difference, which delayed the development of caryopsis, large and loose amyloplast spaces, and reduced amylose content [26]. It was further clarified that the average daily temperature difference and average daily light intensity were the key meteorological factors affecting the canopy microclimate at the panicle and superior and inferior grain chalkiness. The effects of the average daily temperature difference and average daily light intensity on superior grain chalkiness were highest at the mid-filling stage ($r = -0.644$ *~$-0.718$ **), while the effects on inferior grain chalkiness were highest at the early-filling stage ($r = -0.571$ *~$-0.675$ *), complementing and improving the results of previous studies [26–28]. In addition, Xia et al. [36] showed that cumulative temperature at the fruiting stage was significantly and negatively correlated with chalkiness, and that low light intensity reduced the appearance quality of rice [26]. The present study showed that the different times experienced at the early-, mid-, and late-filling stages were the main reasons for the differences in the total temperature difference and effective cumulative temperature in the canopy panicle (Tables 3 and 4). Moreover, the temperature difference and cumulative temperature at the different filling stages have an inconsistent relationship with the chalkiness traits at different grain positions. The total canopy temperature difference was negatively correlated with the chalkiness traits of superior grains ($r = -0.167$~$-0.509$ *) at each filling stage and with the chalkiness traits of inferior grains ($r = -0.581$ *~$-0.675$ *) at the early-filling stage. However, the total canopy temperature difference in the middle and late stages of filling was positively correlated with the chalkiness trait of inferior grains ($r = 0.375$~$0.436$), which would aggravate the chalkiness of inferior grains (Table 6). Except for a negative correlation for inferior grains at the early filling stage, canopy-effective accumulated temperature had a positive correlation in both superior and inferior grains at different filling stages. The results of this study are not consistent with those of Xia et al. [36], which may be caused by regional environmental differences in the planting areas. This study further clarified that the optimal regulation of water and N can improve the canopy temperature and light conditions, which may further promote synergistic interactions among various enzymes, such as sucrose synthase, invertase, adenosine diphosphate pyrophosphorylase, starch synthase, starch branching enzymes, and starch-debranching enzymes within the directly seeded rice grains [37–40].

*4.3. Physiological and Ecological Basis and Regulatory Pathways for Improving Quality and Yield of Directly Seeded Rice under Water–N Interaction*

Water and N availability both play important roles in the regulation of crop yield and quality [4,10,13–16,41]. Currently, the exact manner by which water–fertilizer interaction can fully utilize the water–N coupling effects in rice, and the pathways required to obtain high-quality rice are the focus and hotspot of rice cultivation research [4,41,42]. The results of this study showed that light dry–wet alternation with appropriate N fertilizer application ratio could affect rice chalkiness traits by regulating grain filling characteristics, improving canopy microclimate during the grain filling stage in directly seeded rice. According to our previous results [10,12,13] and the results of this study, the ways to improve the quality of direct seeding rice by water–nitrogen interaction in the early heading stage may concern irrigation and N fertilizer application interactions which can regulate population quality indexes, such as effective tiller number, leaf area index, spatial canopy structure of plant population, root vitality, and material accumulation and translocation. During the grain filling stage, the treatment of water and N fertilizer and canopy microclimate improve grain filling parameters at different grain positions together. The parameters of $T_{max}$, $GR_{max}$, $GR_{mean}$, and $W_{max}$ and their changes directly affected the grain filling and filling quality, which is a direct factor affecting the chalkiness traits. The change in population quality

and grain-filling characteristics parameters at different grain positions causes the canopy microclimate, such as average daily temperature difference, total temperature difference, effective cumulative temperature, and average daily light intensity, to experience different grain filling stages at the panicle, especially the average daily temperature difference and average daily light intensity that indirectly affected two key meteorological factors of rice chalkiness traits. The optimization of grain filling parameters under water–N coupling and the improvement of canopy temperature and light conditions at the panicle may promote synergistic interactions among enzymes, such as sucrose synthase, invertase, adenosine diphosphate pyrophosphorylase, and starch synthase within the seeds of directly seeded rice [26,42], contributing to small and firm amyloplast spaces, which may act as the physiological basis for reducing chalky rice traits [7,32,35,41].

## 5. Conclusions

Water–nitrogen interaction had significant regulatory effects on grain filling characteristics, canopy microclimate, and rice chalkiness (chalky grain rate and chalkiness degree) of directly seeded hybrid *indica* rice. The grain-filling characteristics and canopy microclimate were closely related to rice chalkiness under water–nitrogen interaction conditions. Increasing the maximum and average filling rates of superior and inferior grains during the grain filling stage, and improving the canopy microclimate (average daily temperature difference and average daily light intensity) of inferior grains at the early-filling stage and superior grains at the mid-filling stage are important reasons for the combined effects of water and N coupling to optimize grain filling characteristics and panicle canopy microclimate to reduce rice chalkiness. Under the conditions of a 150 kg hm$^{-2}$ N application rate, flooding irrigation with an N fertilizer application base: tiller: panicle fertilizer ratio of 3:3:4 was appropriate. The best water and N coupling mode for this experiment was the dry–wet alternating irrigation with N fertilizer application in the ratio of 3:5:2. Grain filling characteristics and canopy microclimate can be further optimized to decrease grain chalkiness. In dry alternating irritation, the optimal N fertilizer application base: tiller: panicle fertilizer ratio was 3:5:2, which could provide a reference for production during water shortages.

**Author Contributions:** Y.S. (Yongjian Sun): Conceptualization, Funding acquisition, Methodology, Writing-original draft; Y.W.: Investigation, Writing—original draft; Y.S. (Yuanyuan Sun): Methodology, Software, Writing—original draft; Y.L.: Software, Investigation; C.G.: Data curation; B.L.: Formal analysis; M.X.: Investigation; Z.Y.: Software; F.L.: Data curation, Investigation; J.M.: Conceptualization, Funding acquisition, Supervision, Validation; All authors have read and agreed to the published version of the manuscript.

**Funding:** This work was supported by Sichuan Provincial Science and technology support project (2020YJ0411); the National Key Research and Development Program Foundation of Ministry of Science and Technology of China (2018YFD0301202); the Funding of Academic and Technical Leaders Cultivation Foundation of Sichuan Provincial Human Resources and Social Security Department (2016-183); the Rice Breeding Project Foundation of Sichuan Provincial Science and Technology Department (2021YFYZ0005).

**Institutional Review Board Statement:** Not applicable.

**Informed Consent Statement:** Informed consent was obtained from all subjects involved in the study.

**Data Availability Statement:** The data presented in this study are available on request from the authors.

**Conflicts of Interest:** The authors declare no conflict of interest.

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
