# Peer review of "Effects of Water and Nitrogen on Grain Filling Characteristics, Canopy Microclimate with Chalkiness of Directly Seeded Rice"

_agriculture, doi:10.3390/agriculture12010122_

Round 1

Reviewer 1 Report

This is an interesting manuscript and has the potential to be of interest to agronomists and soil scientists around the world. The manuscript need few minor revisions:

Abstract:

Line 17: Please provide more information on the levels of fertilization (N1, N2, N3, N0)Introduction:It is not known why the authors took up such a research topic? The claim that there is little research in this area is not sufficient reason to conduct research. I suggest that the authors add a research hypothesis.

Materials and Methods:

Line 76: Some data on soil structure would be fine.Line 82: There is little information about agrotechnical side of this experiment - some information should be added. Here more details are needed for tillage, management between harvest and sowing and weed management! Were any herbicides applied or were weeds manually removed? Have fungicides been used?

Author Response

Dear reviewer,

RE: Manuscript agriculture-1348719

We are pleased to submit our revised manuscript (agriculture-1348719) entitled “Relationship of Grain Filling Characteristics and Canopy Microclimate with Chalkiness of Directly Seeded Rice under Water and Nitrogen Interaction”. We appreciate your constructive comments and suggestions.  The manuscript has been modified based on your suggestions. We hope that the revision has sufficiently addressed the concerns and the manuscript is acceptable for publication. Please let me know if you have any questions.

Sincerely,

Dr. Yongjian Sun  

Response to Reviewer 1 Comments

Point 1: Abstract: Line 17: Please provide more information on the levels of fertilization (N1, N2, N3, N0)

Response 1: According to the reviewer’s comment, we added “and three N application strategies under 150 kg ha-1, the application ratio of base: tiller: panicle fertilizer 30%:50%:20% (N1), 30%:30%:40% (N2), 30%:10%:60% (N3), and zero N as the control (N0)” in Abstract: Line 17.

Point 2: Introduction: It is not known why the authors took up such a research topic? The claim that there is little research in this area is not sufficient reason to conduct research. I suggest that the authors add a research hypothesis.

Response 2: According to the reviewer’s comment, we further elaborate what “chalkiness” is in terms of chemical and physical properties, and added a research hypothesis: “Grain chalkiness is a white and opaque part of rice endosperm, which is caused by the insufficient accumulation of starch and protein particles in endosperm [5-6]. It is easy to break during processing, resulting in a significant decrease in milling, appearance, cooking and taste quality of rice [7-9]…how to reduce rice chalkiness by optimized irrigation and nitrogen management…”

Point 3: Materials and Methods: Line 76: Some data on soil structure would be fine.

Response 3: According to the reviewer’s comment,  we added “with the particle composition of sand 45%, clay 35%, silt 20%, 1.25 g·cm-3 bulk density, pH 6.41…” in Line 76.

Point 4: Line 82: There is little information about agrotechnical side of this experiment - some information should be added. Here more details are needed for tillage, management between harvest and sowing and weed management! Were any herbicides applied or were weeds manually removed? Have fungicides been used?

Response 3: According to the reviewer’s comment, we added “Chemical pesticides were used to avoid yield losses and experimental errors due to weeds, insects, and diseases...” in Line 82.

Reviewer 2 Report

Dear Authors,

This is straightforward research. The hypothesis question is not too complicated. These types of studies are conducted in many rice varieties and ecological regions. However, the experimental designs are suitable, the results are statistically processed clearly, and the discussion is appropriate.

The introduction was written in a too general way. Please, write in detail. This will attract and convince readers.

There are a lot of very long sentences in results and discussion,  they are hard for readers. Please re-write (yellow highlight).

Attached is the MS with my comments for your reference. 

Best wishes.

Author Response

Dear reviewer,

RE: Manuscript agriculture-1348719

We are pleased to submit our revised manuscript (agriculture-1348719) entitled “Relationship of Grain Filling Characteristics and Canopy Microclimate with Chalkiness of Directly Seeded Rice under Water and Nitrogen Interaction”. We appreciate your constructive comments and suggestions.  The manuscript has been modified based on your suggestions. We hope that the revision has sufficiently addressed the concerns and the manuscript is acceptable for publication. Please let me know if you have any questions.

Sincerely,

Dr. Yongjian Sun 

Response to Reviewer 2 Comments

Point 1: The introduction was written in a too general way. Please, write in detail. This will attract and convince readers.

Response 1: According to the yellow mark of the reviewer, we carefully revise the introduction, as detailed in the revised manuscript, thank you very much.

Point 2: There are a lot of very long sentences in results and discussion, they are hard for readers. Please re-write (yellow highlight).

Response 2: According to the yellow mark of the reviewer, we carefully re-write each long sentence marked (yellow highlight) by the reviewer, as detailed in the revised manuscript.

Reviewer 3 Report

A comprehensive study unravelling complex interactions to show how water and N interact to affect rice grain filling and quality. However, understandability needs to be improved by reducing repetition and, particularly in the Discussion, shortening some extraordinarily long sentences. The readability needs to be improved.

Title

Suggested alternative: Effects of Water and Nitrogen on Grain Filling Characteristics, Canopy Microclimate with Chalkiness of Directly Seeded Rice

Abstract

L13-15. More or less repeats the title. Need a more general statement of aim, along the lines of “how to reduce chalkiness of rice grain”.

L21. The term “friendly canopy microclimate” unclear. Something to do with leafiness or excessive vegetative growth.

L23. “superior and inferior grains” and L25-27also unclear.

Abstract is difficult to understand without having first read the entire paper. Abstract should be understandable by itself and needs to be rewritten.

Introduction

L40. Need to elaborate what “chalkiness” is in terms of chemical and physical properties, i.e. how and why does it affect quality?

L47. “constrain”? Do you mean “interact”?

L46-70. Although this is understandable as is it is repetitive and could be expressed in a more concise manner.

Materials and Methods

L77-78. Need to refer to what methods used to measure these parameters.

L103-4. Perhaps mention pest and disease control measures, and indicate if the crops were free from biotic stresses, as any such stresses would inevitably affect the results obtained.

L160-2. Rather than simply giving references please further elaborate on what chalkiness actually is and how it was measured in this study.

Results

L170. Should be “grain-filling rate”

L278. Should be “temperature difference”

L326-32. Break up this long sentence – difficult to follow as is.

Discussion

L393. “irrational irrigation dynamics” – meaning unclear.

L450-62. Extremely long sentence, difficult to understand. Please break up into several sentences. Similarly for L462-8 and L470-8 and L484-90 and L500-8. Generally, in the Discussion sentences are too long and readability and understandability could be improved by shortening them.

L483. “inferior grain filling wall” – unclear.

Conclusion

Shorter sentences also warranted here also.

Author Response

Dear reviewer,

RE: Manuscript agriculture-1348719

We are pleased to submit our revised manuscript (agriculture-1348719) entitled “Relationship of Grain Filling Characteristics and Canopy Microclimate with Chalkiness of Directly Seeded Rice under Water and Nitrogen Interaction”. We appreciate your constructive comments and suggestions.  The manuscript has been modified based on your suggestions. We hope that the revision has sufficiently addressed the concerns and the manuscript is acceptable for publication. Please let me know if you have any questions.

Sincerely,

Dr. Yongjian Sun 

Response to Reviewer 3 Comments

Point 1: Title: Suggested alternative: Effects of Water and Nitrogen on Grain Filling Characteristics, Canopy Microclimate with Chalkiness of Directly Seeded Rice

Response 1: According to the reviewer’s suggestion, modify the original title to “Effects of Water and Nitrogen on Grain Filling Characteristics, Canopy Microclimate with Chalkiness of Directly Seeded Rice”

Point 2: Abstract: L13-15. More or less repeats the title. Need a more general statement of aim, along the lines of “how to reduce chalkiness of rice grain”.

Response 2: According to the reviewer’s comment, we re-write L13-15 and reduce repetition of discussions with topics, “In order to determine how to reduce the chalkiness of rice grains through irrigation modes and nitrogen (N) fertilizer management.”

Point 3: Abstract: L21. The term “friendly canopy microclimate” unclear. Something to do with leafiness or excessive vegetative growth.

Response 3: According to the reviewer’s comment, we have used “canopy microclimate parameters” instead of “friendly canopy microclimate” in L21.

Point 4:Abstract: L23. “superior and inferior grains” and L25-27also unclear. Abstract is difficult to understand without having first read the entire paper. Abstract should be understandable by itself and needs to be rewritten.

Response 4: According to the reviewer’s comment, we have used “superior grains of the primary branches and inferior grains of the secondary branches” instead of “superior and inferior grains” in L23.

Point 5: Introduction:L40. Need to elaborate what “chalkiness” is in terms of chemical and physical properties, i.e. how and why does it affect quality?

Response 5: According to the reviewer’s comment, we added “Grain chalkiness is a white and opaque part of rice endosperm, which is caused by the insufficient accumulation of starch and protein particles in endosperm [5-6]. It is easy to break during processing, resulting in a significant decrease in milling, appearance, cooking and taste quality of rice [7-9]…”

Point 6: Introduction:L47. “constrain”? Do you mean “interact”?

Response 6: According to the reviewer’s comment, we have used “interact” instead of “constrain” in L47.

Point 7: Introduction: L46-70. Although this is understandable as is it is repetitive and could be expressed in a more concise manner.

Response 7: According to the reviewer’s comment, we carefully revise the introduction, as detailed in the revised manuscript, thank you very much.

Point 8: Materials and Methods:L77-78. Need to refer to what methods used to measure these parameters.

Response 8: According to the reviewer’s comment, we added “25.6 g·kg-1 organic matter (K2Cr2O7-volumetric method), 1.71 g·kg-1 total N (Kjeldahl method, FOSS-8400, Sweden), 25.2 mg·kg-1 available phosphorus (Olsen method), and 154.7 mg·kg-1 available potassium (flame spectrometry after NH4OAc extraction) of composite topsoil samples (0-20 cm)…”

Point 9: Materials and Methods:L103-4. Perhaps mention pest and disease control measures, and indicate if the crops were free from biotic stresses, as any such stresses would inevitably affect the results obtained.

Response 9: According to the reviewer’s comment, we added “Chemical pesticides were used to avoid yield losses and experimental errors due to weeds, insects, and diseases...”

Point 10: Materials and Methods: L160-2. Rather than simply giving references please further elaborate on what chalkiness actually is and how it was measured in this study.

Response 10: According to the reviewer’s comment, we added “the chalky grain rate and chalkiness of superior and inferior grains were determined according to the methods described by Yoshioka et al [21] and Chen et al [22] with some modification. The samples of superior and inferior grain each at 50 g were taken from each plot. Rough rice was de-husked using a husker (JLGJ 4.5, Taizhou Cereal and Oil Instrument Co., Ltd., Zhejiang, China). From each sample, 20g brown rice was taken for milling using a Miller (JNM-â…¢, China Grain Reserve Chengdu Research Institute Co., Ltd., Chengdu, China), and then head rice was divided from broken rice. A digital image was created by placing head rice of each sample on the scanner and the appearance quality indicators of chalky grain rate and chalkiness degree was analyzed by Image analyzing software (JMWT-12, Beijing Dongfu Jiuheng Instrument Co., Ltd., China and Japan Satake Co., Ltd., Japan)” in L160-2.

Point 11: Results: L170. Should be “grain-filling rate”

Response 11: According to the reviewer’s comment, we have used “grain-filling rate” instead of “gain filling rate” in L170.

Point 12: Results:L278. Should be “temperature difference”

Response 12: According to the reviewer’s comment, we have used “temperature difference” instead of “temperature different” in L278.

Point 13: Results:L326-32. Break up this long sentence – difficult to follow as is.

Response 13: According to the reviewer’s comment, we re-write the long sentence “Furthermore, the relationship between the MGR of superior and inferior grain and the canopy microclimate indexes were closely related to chalky grain rate and chalkiness degree at the early, middle, and late-filling stages (Table 6). Compared with the early and last-filling stages, the MGR of superior and inferior grains at the mid-filling stage was significantly negatively correlated with the chalkiness traits of superior and inferior grains (r=-0.676*~-0.750**). The highest correlation coefficient was observed at the mid-filling stage.”

Point 14: Discussion:L393. “irrational irrigation dynamics” – meaning unclear.

Response 14: According to the reviewer’s comment, we have used “which could lead to unreasonable grain filling: prolonged D, decreased GRmax, and GRmean” instead of “irrational irrigation dynamics” in L393.

Point 15: Discussion: L450-62. Extremely long sentence, difficult to understand. Please break up into several sentences. Similarly for L462-8 and L470-8 and L484-90 and L500-8. Generally, in the Discussion sentences are too long and readability and understandability could be improved by shortening them.

Response 15: According to the reviewer’s comment, we re-write the long sentences: “The findings of the present study showed that there was a significant mutual regulatory effect of water and N treatments on canopy microclimate (Tables 3 and 4). There were significant differences in the canopy microclimate at the early, middle, and late-filling stages of superior and inferior grains under different water and N treatments. The effect of N fertilizer on canopy microclimate was significantly higher than that of irrigation mode. With the proportion of delayed N fertilizer application increased, the average daily temperature difference and average daily light intensity showed a decreasing trend in different degrees. The possible reasons for the increase of rice chalkiness caused by the increase in the percentage of postponing N topdressing are the increase of leaf area index, closure and humidity of the canopy [12, 14], the decrease of light transmission [12, 26], the night temperature could not be decreased quickly [27-28], resulting in an increase in the average daily temperature difference, which delayed the development of caryopsis, large and loose amyloplast spaces, and reduced amylose content [26].” In L450-62.

Similarly for L462-8 and L470-8 and L484-90 and L500-8. We carefully re-write each long sentence marked by the reviewer, as detailed in the revised manuscript.

Point 16: Discussion: L483. “inferior grain filling wall” – unclear

Response 16: According to the reviewer’s comment, we have used “inferior grain filling stage” instead of “inferior grain filling wall” in L483.

Point 17: Conclusion: Shorter sentences also warranted here also.

Response 17: According to the reviewer’s comment, we shorted the long sentence in part of Conclusion, as detailed in the revised manuscript.
